# Utility of Leaf Area Index for Monitoring Phenology of Russian Forests

**Nikolay V. Shabanov** [1,2,*], **Vyacheslav A. Egorov** [1], **Tatiana S. Miklashevich** [1], **Ekaterina A. Stytsenko** [1] and **Sergey A. Bartalev** [1]

[1] Space Research Institute of the Russian Academy of Sciences, 117997 Moscow, Russia; egorov@d902.iki.rssi.ru (V.A.E.); limires@d902.iki.rssi.ru (T.S.M.); ekaterina_stytsenko@d902.iki.rssi.ru (E.A.S.); bartalev@d902.iki.rssi.ru (S.A.B.)

[2] Faculty of Space Research, Lomonosow Moscow State University, 119234 Moscow, Russia

[*] Correspondence: nikolay.shabanov@d902.iki.rssi.ru

**Abstract:** Retrievals of land surface phenology metrics depend on the choice of base variables selected to quantify the seasonal "greenness" profile of vegetation. Commonly used variables are vegetation indices, which curry signal not only from vegetation but also from the background of sparse foliage, they saturate over the dense foliage and are also affected by sensor bandwidth, calibration, and illumination/view geometry, thus introducing bias in the estimation of phenometrics. In this study we have intercompared the utility of LAI and other biophysical variables (FPAR) and radiometric parameters (NDVI and EVI2) for phenometrics retrievals. This study was implemented based on MODIS products at a resolution of 230 m over the entire extent of Russian forests. Free from artifacts of radiometric parameters, LAI exhibits a better utilization of its dynamic range during the course of seasonal variations and better sensitivity to the actual foliage "greenness" changes and its dependence on forest species. LAI-based retrievals feature a more conservative estimate of the duration of the growing season, including late spring (9.3 days) and earlier fall (8.9 days), compared to those retrieved using EVI2. In this study, we have tabulated typical values of the key phenometrics of 12 species in Russian forests. We have also demonstrated the presence of the latitudinal dependence of phenometrics over the extent of Russian forests.

**Keywords:** phenology; leaf area index; seasonality of broadleaf and needleleaf forests; Russian forests





## 1. Introduction

Land surface phenology metrics are statistical measures of the seasonal dynamics of vegetation "greenness". They are also sensitive indicators of state and change in various ecosystem processes involving carbon, water, energy, and nutrient cycling [1,2]. Phenometrics are required in many applications for land monitoring, climate and carbon cycle studies. In particular, phenology data have been used for land cover/land use/land cover change mapping [3,4], monitoring crop type/crop yield/crop management practices [5,6], assessing vegetation response to climate change [7,8], and others. Phenometrics are an essential part of the parametrization of land surface models [2] and terrestrial carbon cycle models [9]. Phenology has been identified as a critical parameter for global climate change research according to the IPCC 6th assessment report https://www.ipcc.ch/report/ar6/wg2/ (accessed on 13 November 2023).

Over the past few decades, various phenological studies were implemented based on remote sensing observations on regional and global scales. Due to restrictions on repeat frequency, most phenological products were implemented using data from polar-orbiting moderate resolution imagers (daily global data at a few km resolutions), such as AVHRR [10], SPOT-VGT [11], MERIS [12], MODIS [13], and VIIRS [14] data. With the recent availability of several high-resolution (10–30 m) sensors on the orbit (Landsat, Sentinel-2A, and -2B), phenological products at high resolution were finally generated over

the continental scale of North America [15]. Another promising direction of research is to utilize data from geostationary satellites, such as SEVIRI [16] and AHI [17], to minimize the effect of cloud cover due to the high frequency of observations (hourly data).

Statistical properties of phenometrics depend on the choice of base variables, seasonal profiles of which are analyzed in retrievals. Most current phenological products are retrieved based on the Vegetation Indices (VI)-radiometric variables, which curry signal not only from vegetation "greenness" but also from background for sparse foliage. They are affected by sensor bandwidth and calibration and illumination/view geometry (Bidirectional Reflectance Distribution Function, BRDF effect) and saturate over dense foliage [18]. Among the most widely used base variables are the Normalized Difference Vegetation Index (NDVI) and Enhanced Vegetation Index (EVI). Some biophysical parameters, such as Leaf Area Index (LAI), are free from all the above artifacts. Others only partially remove them: for instance, Fraction of Photosynthetically Active Radiation Absorbed by Vegetation (FPAR) can be retrieved for fixed geometry and background reflectance but still saturates [19,20]. Biophysical parameters require significant efforts to retrieve and involve modeling approximations. Given the costs, a practical implementation question arises: is there an advantage of using biophysical variables for phenometrics retrievals? Recent studies give a positive answer: according to [21], Net Primary Productivity (NPP) or LAI helps to improve the accuracy of the estimation of phenological phases over Pan-Arctic regions because, under the condition of the short growing season and low availability of cloud- and snow-free observations, those parameters are less prone to disturbances.

In this study, we have attempted to both conceptually understand and practically assess the utility of LAI with respect to other biophysical (FPAR) and radiometric (NDVI and EVI2) parameters for retrieval of phenometrics over forests. This research has been implemented using MODIS data over the entire extent of Russian forests, which exhibit a wide range of foliage density and seasonal dynamics. Note that while several global (or Pan-Arctic) phenological products exist, we did not find a focused in-depth assessment of the phenology of this region.

This paper is organized as follows. Section 2 describes the remote sensing products utilized in this research and provides a background for the conceptual understanding of the difference in seasonal variations in the base variables selected for this study: NDVI, FPAR, EVI2, and LAI. We also briefly highlight features of our phenometrics retrieval algorithm. Section 3 reports on the results of this study, which includes the analysis of data coverage limitations, comparison of dynamic properties of base variables, analysis of retrieved phenometrics, and sensitivity analysis.

## 2. Materials and Methods

In this study, we have implemented retrievals of the forest phenometrics from the annual time series of daily NDVI, EVI2, FPAR, and LAI products generated from Terra MODIS observations. Regular production of MODIS LAI time series is performed at the Space Research Institute of the Russian Academy of Sciences (IKI). For this study, we have also implemented the same chain of processing for VIs and FPAR data. Those products are described in Section 2.1. Section 2.2 describes NASA MODIS MCD12Q2 product utilized as a reference for phenometrics retrieval methodology and also for product intercomparison purposes. Appendix A describes IKI MODIS forest species product referenced to quantify dependence of phenometrics.

### 2.1. IKI MODIS NDVI, EVI2, FPAR, and LAI Products

The IKI MODIS LAI is a cloud- and snow-screened daily composited and interpolated product at 230 m resolution from January 2001 to the present [22]. LAI (dimensionless) is defined as one-sided green leaf area per unit ground area in the broadleaf canopies and as one-half the total needle surface area per unit ground area in coniferous canopies [23]. In this study, we utilize total forest LAI; in other words, we did not separate foliage into overstory and understory [22]. The daily LAI retrieval algorithm is an enhanced version of the NASA

MODIS Collection 6 MOD15 algorithm https://lpdaac.usgs.gov/documents/624/MOD1
5_User_Guide_V6.pdf (accessed on 13 November 2023). LAI is retrieved from the MODIS
Red and NIR channel data (Collection 6) using the Stochastic RT model simulations [24] for
a range of canopies, including forests. Retrievals are performed regardless of cloud/snow
conditions. Next, daily LAI is screened with cloud/snow mask, and (if available) multiple
swath data are composited. In the next step, temporal interpolation of annual time series
is performed. Algorithm implements data correction and recovery based on second-
order polynomial approximation in a sliding temporal window of variable size (to keep
10 valid data points) [25]. The objective of this step is outlier removals, gap filling and
general seasonal curve smoothing. Output product is stored at a daily temporal resolution.
Compared to the original NASA version, the following enhancements were made to the
LAI algorithm: (1) implemented new RT simulations using the latest enhancements to the
Stochastic RT model, including extension of the range and increase in discretization of the
Solar Zenith Angle (SZA) (15–75°), separation of simulations for needleleaf forests between
evergreen and deciduous classes; (2) incorporated new ancillary input- an annual 8-biome
IKI MODIS landcover at 230 m resolution; (3) performed screening of artifacts (stripes) of
MODIS Red channel data (c.f. Appendix B). Accuracy assessment of the products utilized
in this study is summarized in the Supplementary Materials document accompanying this
article. In this study, we utilized IKI products for 2020 over all of Russia.

FPAR (dimensionless) product is also retrieved using the daily IKI MODIS LAI algo-
rithm according to the original methodology of the NASA MODIS MOD15 algorithm [19].
Mathematically, FPAR is defined as

$$\text{FPAR} = \int_{400nm}^{700nm} \text{a}(\lambda) \cdot \text{e}(\lambda) \text{d}\lambda \tag{1}$$

where $\lambda$ is a wavelength, a($\lambda$) is spectral canopy absorptance at direct solar illumination in
the nadir direction, and e($\lambda$) is the normalized Plank function, approximating incident solar
radiation. The integral is taken over part of the solar spectrum [400–700 nm] utilized via
vegetation in the process of photosynthesis. Given daily FPAR data, the same processing as
for LAI is applied: cloud/snow masking, daily compositing, and daily interpolation. The
IKI MODIS FPAR product is in the evaluation phase currently and has not been validated
with ground measurement by the IKI team. However, validation of the baseline NASA
MOD15 FPAR product is ongoing [16].

MODIS channel data (Red and NIR) used to retrieve IKI MODIS LAI products were
also saved in the daily retrievals output file and were used to construct various VIs,
including NDVI and two-channel EVI (EVI2), defined as

$$\text{NDVI} \equiv (\text{N} - \text{R})/(\text{N} + \text{R}), \tag{2}$$

$$\text{EVI2} \equiv \text{G} \cdot (\text{N} - \text{R})/(\text{N} + \text{C} \cdot \text{R} + 1), \tag{3}$$

where N and R are channel data at NIR and Red spectral bands; values of parameters of
EVI2 were optimized for MODIS sensor, namely G = 2.5, C = 2.4 [26]. Given daily NDVI and
EVI, the same used for LAI processing was applied to construct daily interpolated products.

### 2.2. NASA MODIS MCD12Q2 Phenology Product

The NASA MODIS MCD12Q2 phenology product https://modis-land.gsfc.nasa.g
ov/pdf/MCD12Q2_Collection6_UserGuide.pdf (accessed on 13 November 2023) is an
annual product at resolution of 500 m. It provides estimates of phenological metrics (c.f.
Table 1) based on EVI2. Inputs to the algorithm are three years of time series of 16-day
composited product MODIS NBAR EVI2. EVI2 was calculated not directly from MODIS
channel data affected by varying illumination/observation geometry but rather from the
normalized reflectances available from the NASA MODIS MCD43 product: Nadir BRDF
Adjusted Reflectance (NBAR). NBAR channel data provide an estimate of reflectance at
nadir, given solar illumination at noon local time. Given the fact that SZA is not fixed,

this is an important, but only a partial normalization of channel data. Also, cost of this normalization is the use of long-time composting (16 days), which is especially undesirable during phenological phases with rapid changes (spring and fall). Further processing, cubical spline fitting of NBAR EVI2 helps to fill gaps and increase temporal resolution but still at the cost of suppressing short-term (non-linear) changes in VI due to non-monotonical weather forcing.

**Table 1.** Key phenometrics retrieved using NASA MCD12Q2 and IKI phenology algorithms. Here, "base variable" denotes EVI2 (for the MCD12Q2 product) or LAI (for the IKI phenology product). Phenometrics marked by (*) were introduced in this study and not available in the MCD12Q2 product.

| Phenometrics | Definition |
| --- | --- |
| greenup | Date when base variable first crosses 15% of amplitude of its variations |
| mid-greening | Date when base variable first crosses 50% of amplitude of its variations |
| maturity | Date when base variable first crosses 90% of amplitude of its variations |
| maximum | Date when seasonal maximum is achieved |
| senescence | Date when base variable last crosses 90% of amplitude of its variations |
| mid-browning | Date when base variable last crosses 50% of amplitude of its variations |
| dormancy | Date when base variable last crosses 15% of amplitude of its variations |
| minimum | Minimum value of the base variable |
| duration * | Difference between mid-browning and mid-greening dates |
| greening spread * | Difference between maturity and greenup dates |
| browning spread * | Difference between formancy and denescence dates |
| amplitude | Difference between maximum and minimum of base variable |
| integral | Integral of base variable from greenup to dormancy |

## 2.3. Empirical and Theoretical Background on Seasonal Variation in NDVI, EVI2, and LAI

As previously stated, statistical properties of phenometrics depend on the choice of base variable, seasonal profile of which is analyzed. To conceptually understand reasons behind this fact, consider sample seasonal profiles of the IKI MODIS NDVI, EVI2, and LAI products over Spasskaya Pad site in Yakutia (62.25500°N, 129.61880°E), as shown in Figure 1a. This is a FLUXNET site https://fluxnet.org/ (accessed on 13 November 2023), larch being a dominant species. As variables have different dynamic ranges, they need to be scaled in order to perform valid comparison of seasonal profiles. We matched seasonal maximum of variables by scaling each by corresponding seasonal maximum values, $NDVI_{max}$, $EVI2_{max}$, and $LAI_{max}$. We did not switch from variables to the corresponding amplitudes because we want to work with full dynamic range, which may include large and varying minimum in winter time for VIs carrying information both on vegetation and background. Comparing seasonal course of scaled variables (Figure 1a), one can notice that the growing season duration decreases from NDVI to EVI2 to LAI.

This is a general rule for vegetation canopy. To demonstrate this, we have utilized radiative transfer simulations underlying the IKI MODIS LAI algorithm in the form of dependencies between channel data (Red and NIR) and LAI, from which the relationship between NDVI/EVI2 and LAI can be calculated. Simulations were performed for deciduous needleleaf forests, SZA = 37.5°, VZA = 9.5°, RA = 180°, soil hemispherical reflectance = 0.09. Figure 1b shows dependence of the normalized VIs and LAI from LAI. For given LAI value, scaled VIs are always higher than scaled LAI. Thus, concave form of the relationship between VIs and LAI explains wider growing season duration of VIs compared to that of LAI. In turn, non-linear relationship between VIs and LAI comes from the non-linear relationship between channel reflectances and LAI. The latter is just an expression of

saturation of channels, that is, decrease in sensitivity of channels to LAI ($\partial X/\partial LAI$, X is a channel) with increase in foliage density.

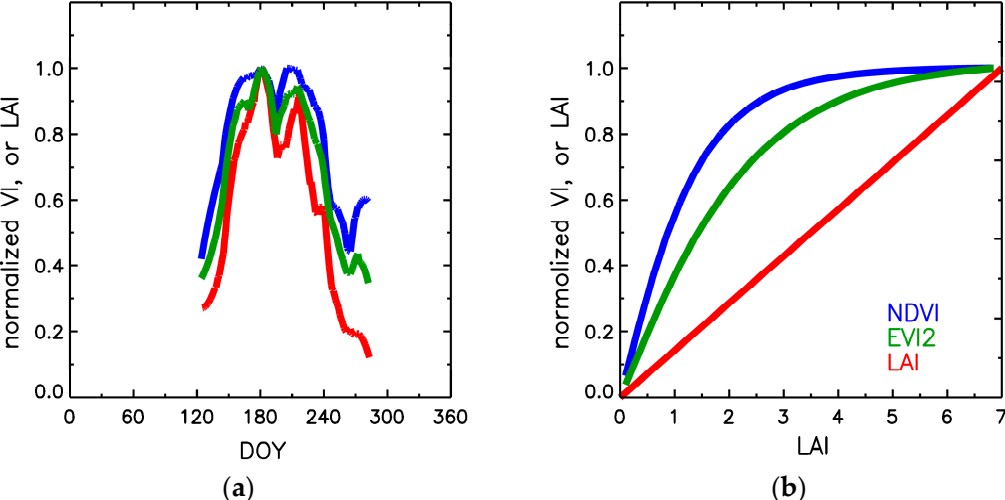

**Figure 1.** Analyzing the differences and their causes of seasonal variations of NDVI, EVI2, and LAI. Panel (**a**) shows seasonal profiles of VIs and LAI, scaled by corresponding seasonal maximum for Spasskaya Pad site in Yakutia, a larch being a dominant species (deciduous needleleaf forest class). Panel (**b**) shows relationship between scaled NDVI/EVI2 and LAI based on the radiative transfer simulations underlying IKI MODIS LAI algorithm for the same forest class.

Further insights into the relationship between variations of VIs and LAI can be obtained by inspecting VIs/LAI isolines in the Red/NIR spectral space (Figure 2). Isolines of VIs/LAI are defined as lines in the Red-NIR spectral space along which value of the variable remains unchanged. Equations for isolines of NDVI and EVI2 can be easily derived from the definition of those indices by setting index to constant and inverting for the relationship NIR (Red):

$$\text{NDVI} = \text{const} = A \rightarrow \quad N = (1+A)/(1-A))R, \tag{4}$$

$$\text{EVI2} = \text{const} = B \rightarrow N = (G+C{\cdot}B)/(G-B)R + B/(G-B), \tag{5}$$

where A and B are constant values defining VIs isolines; parameters for EVI2 are defined in Equation (3). Equation for LAI isolines cannot be derived analytically but rather numerically based on the radiative transfer simulations (the same parameters were used as for Figure 1b, except soil reflectance was allowed to vary in the full range (0.01–0.15)). LAI isolines start from soil line (when LAI = 0)—ray starting from Red-NIR space origin. As LAI is increasing, we obtain set of isolines with increasingly steeper inclination and also shifted from the origin. Set of NDVI isolines are set of rays starting from the origin. They only match LAI isolines in sense that they become steeper with foliage density increases. Definition of NDVI is not parametric, and thus, the fundamental limitation that NDVI isolines always originate from zero, while LAI isolines deviate from the origin, i.e., cannot be fixed. In this sense, EVI2 matches LAI better; isolines correspond to set of lines getting steeper with increase in foliage density and not originate from zero. To attain a close match between EVI2 and LAI isolines, one needs to constantly change parameters of EVI2. For a fixed set (like in case of the MODIS EVI2 product), this is not possible. To summarize, EVI2 is better than NDVI approximates LAI, as EVI2 isolines are more flexible to approximate those of LAI (Figure 2), which ultimately results in a better match of seasonal variations (Figure 1a).

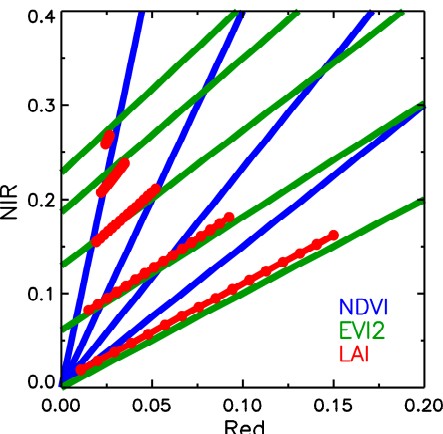

**Figure 2.** Quantifying variations of foliage "greenness" in the Red-NIR spectral space using NDVI, EVI2, and LAI isolines. Isolines are curves along which corresponding variable is constant. True measure of "greenness" is specified by LAI, while NDVI and EVI2 are proxies.

### 2.4. Retrievals of Phenometrics from the Seasonal LAI Profiles

We've adopted methodology for retrieval phenometrics of Collection 6 NASA MODIS MCD12Q2 algorithm https://modis-land.gsfc.nasa.gov/pdf/MCD12Q2_Collection6_UserGuide.pdf (accessed on 13 November 2023), [15] but applied those to LAI instead of EVI2. We have made the following changes to the processing algorithm. Instead of processing three years of data (current and two adjacent years), we have implemented retrievals based only on current year seasonal profile. Due to the limited amount of snow/cloud-free data during winter in Russian forests, we cannot establish reliable link between adjacent years. Also, we assume there exist only a single seasonal peak of LAI over Russian forests. Finally, our temporal interpolation is based on short-term data fitting to quadratic polynomials over moving window of 10 measurements rather than cubic spline fitting over whole annual cycle in attempt to preserve natural variability of foliage development. Otherwise, processing the seasonal profile is the same to estimate set of phenometrics according to their definition (c.f. Table 1). Note, in addition to standard phenometrics defined in the MCD12Q2 product, we introduced duration, greening spread, and browning spread, which can be calculated from the standard set of variables.

## 3. Results

### 3.1. MODIS Data Coverage Limitations

Forests of Russia occupy high northern latitudes (40°–80°N); some of them (such as larch forests) are located in the central part of a large continent with harsh weather conditions, all leading to a prolonged vegetation dormancy period under cloudy/snow conditions. In fact, limited clear sky/snow-free remote sensing observations and low SZA were identified as major problems for accurate retrievals of phenometrics in the Pan-Arctic region [21]. Analysis of the time series of the IKI MODIS LAI product indicates that during winter (DOY < 60 and DOY > 330, that is, January–February and December, where DOY stands for Day Of the Year), data coverage is very low (<5%) and concentrated mostly at the southern regions of the Russian forests (cf. Figure 3). In contrast, during the growing season (DOYs 160–270, that is, June–September), data availability reaches nearly 100%. Thus, while a seasonal maximum of VI/ FPAR/LAI can be reliably estimated, the seasonal minimum during the spring/fall may be thought of as uncertain, especially for high northern latitudes. Seasonal variations of LAI will be discussed later, but at this point, we can state that the average date of reaching min value (LAI < 1) is DOY = 126 for spring and DOY = 281 for fall, marked as the brown interval in Figure 3a. Thus, the seasonal course of LAI closely follows the course of data availability: cleared from snow and winter cloudiness, foliage develops. Still, especially for evergreen species in winter, LAI values may be of interest in terms of phenology. However, during the winter, the temperature in the region drops

below the biological threshold of +5 °C, at which even evergreen species absorb PAR but stop photosynthesis as stomata are forced to close [27]. Thus, this period is out of the scope of our study, as we are focused on phenometrics retrievals for green photosynthesizing vegetation. Consistency between temperature and data availability can be analyzed in the future according to the methodology of [28].

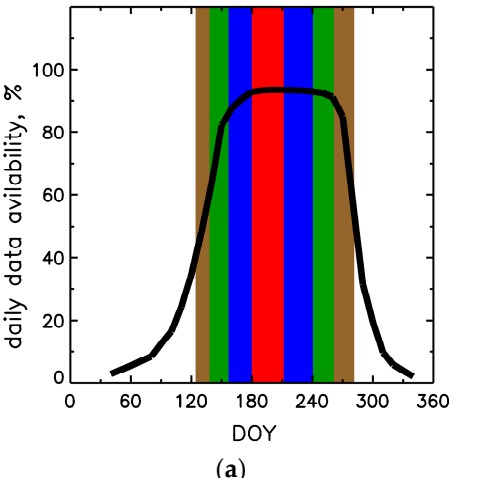 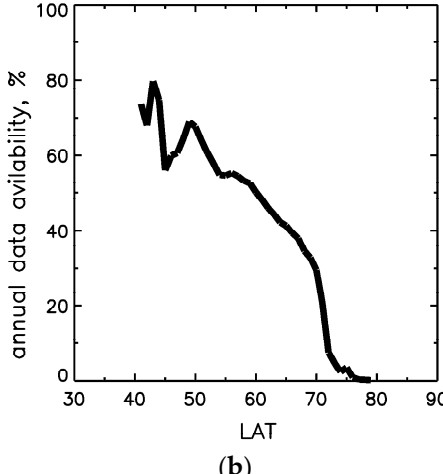

(**a**) (**b**)

**Figure 3.** Limitations of the spatio-temporal coverage of the IKI MODIS LAI product over Russian forests. Panel (**a**) shows annual course of data availability and ratio of number of pixels with valid LAI retrievals to total forest pixels for a given DOY. To match data availability with seasonal LAI variations also are shown intervals of DOYs when LAI reaches (or exceeds) the following portions of amplitude of variations: 90% over DOYs 180–211 shown in red ■, 50% over DOYs 157–240 in blue ■, and 15% over DOYs 138–261 in green ■. At the boundaries of brown ■ interval DOYs 126–281, LAI reaches minimum in spring and fall. Panel (**b**) shows latitudinal distribution of annual data availability, the ratio of annual sum of valid LAI pixels to total forest pixels in a given latitudinal band.

*3.2. Comparison of Dynamic Properties of NDVI, EVI2, FPAR and LAI*

Which variable is most suitable for phenometrics retrievals? For comparison, we will utilize two radiometric (NDVI and EVI2) and two biophysical (FPAR and LAI). We've demonstrated the theoretical advantages of LAI in the Section 2.3 as an index that measures the true "greenness" of foliage, while NDVI and EVI2 are only proxies and are corrupted by various artifacts. Still, questions remain, given the theoretical advantages of LAI, do those translate into superior dynamic properties of seasonal variations?

Figure 4 demonstrates a seasonal sequence of histograms of NDVI, FPAR, EVI2, and LAI calculated over the full extent of the Russian forest for DOYs 120, 170, 200, 230, and 280, covering major phases of seasonal changes from greening to browning. Comparing the dynamics of NDVI and FPAR histograms, one can assert that variables have a very similar performance from the seasonal cycle: average values reaching ~0.5 at the beginning (DOY~126) and the end (DOY~281) and 0.9 at the peak of the growing season (DOY~195). This comes from the fact that FPAR and NDVI have close to linear relationships affected by soil and vegetation variations, geometry of observations, etc. [20]. During the summer peak, both variables are strongly affected by saturation (values are concentrated at a narrow peak). At the beginning and the end of the growing season, variables are far from reaching the lower limit (zero), in fact, they reach only the middle of the dynamic range.

Thus, due to saturation and the inability to exploit full dynamic range, NDVI and FPAR are not the best candidates for phenology monitoring. Next, the seasonal sequence of EVI2 histograms is considered. In terms of dynamic properties, EVI2 shows some improvements: during the summer period, histograms are quite stable (DOYs 170–230), the beginning/end values (DOYs 120 and 280) are closer to the lower end of the variations (~0.2). However, summer values are lower than that of NDVI, on average ~0.5, thus

suppressing the utilization of the full dynamic range. Also, in terms of saturation, EVI2 does not show substantial improvement over NDVI; while the seasonal peak moved to lower values, the spread is quite similar to that of NDVI. Finally, consider the seasonal variations in LAI histograms. This variable addresses the issues found for other variables: seasonal beginning/end values are at the lower end of variations, and in the summer, the nearly full dynamic range of variables is utilized.

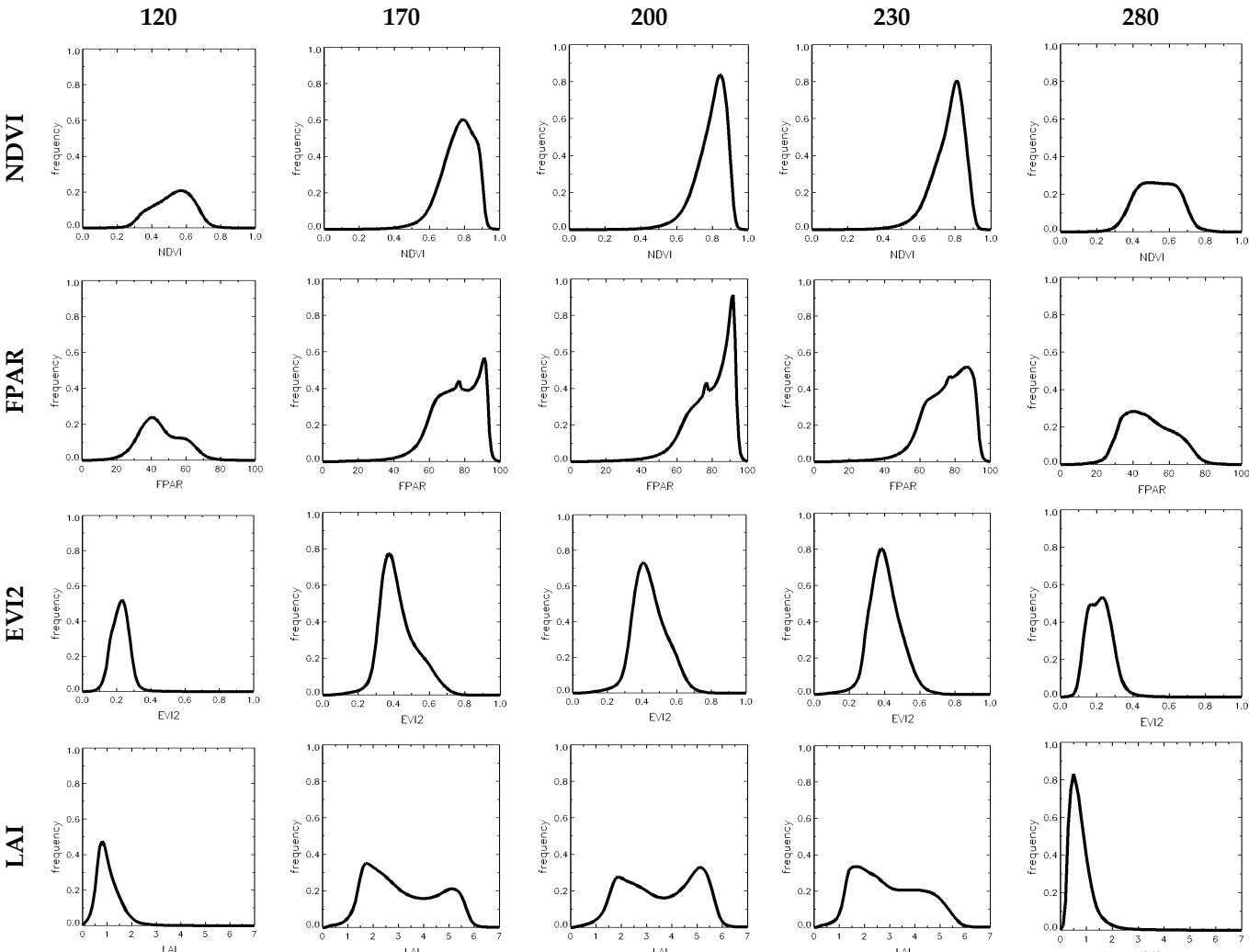

**Figure 4.** Seasonal sequence of distributions of NDVI, FPAR, EVI2, and LAI over Russian forests. Row specifies variable (NDVI, FPAR, EVI2, or LAI), while column indicates DOY, at which histogram was calculated.

Now, the seasonal profiles of NDVI, EVI2, and LAI averaged over DNF, ENF, and DBF (Deciduous Needleleaf Forests, Evergreen Needleleaf Forests, and Deciduous Broadleaf Forests) are compared, as shown in Figure 5. More details on seasonal variations by forest species are given for LAI in Figure 6. The following should be noted. All three parameters indicate consistent variations, as VIs being proxies of LAI to characterize "greenness". Still, VIs exhibit deficiencies over LAI (c.f. Figure 5): (a) outside the growing season stay high, so it is hard to identify if the canopy has photosynthesizing foliage, and (b) during the growing season suppresses variation in greenness between classes. Large values of NDVI (~0.4) outside of the growing season over DBF in North America are also reported in [29].

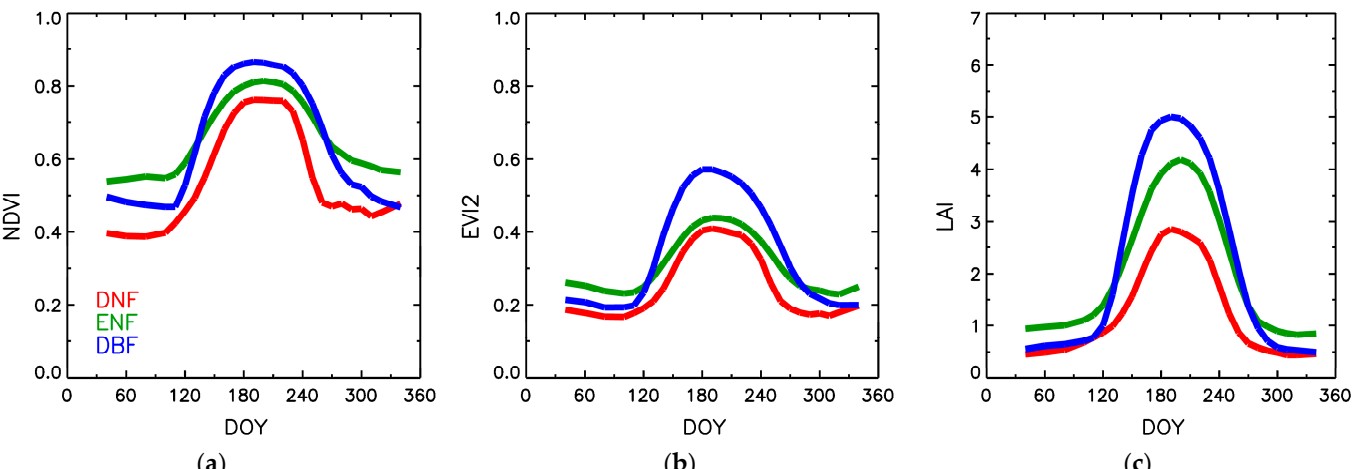

**Figure 5.** Seasonal course of NDVI (**a**), EVI2 (**b**), and LAI (**c**) as function of classes of Russian forests. Each curve was created by averaging NDVI/EVI2/LAI values for particular DOY over all pixels occupied by a particular class.

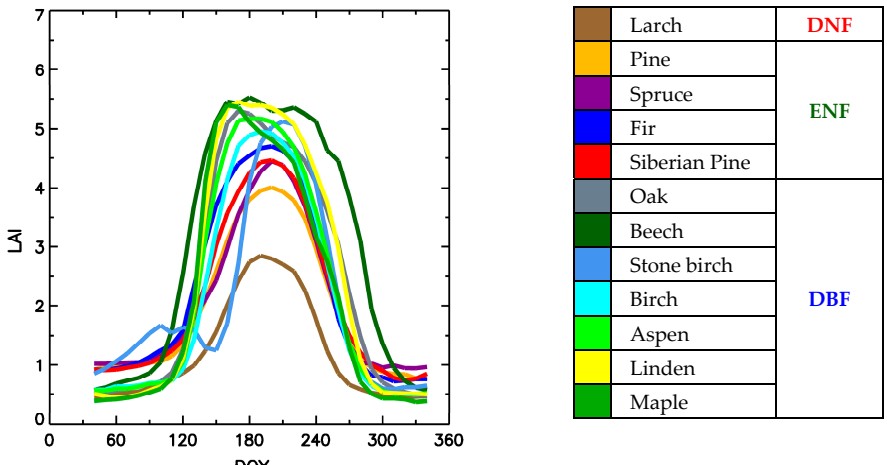

**Figure 6.** Seasonal course of LAI as function of species of Russian forests. Each curve corresponds to average LAI values, calculated for particular DOY over all pixels occupied by a particular species.

Those deficiencies are again due to the fact that VIs are only proxies to LAI. For sparse foliage, they are affected by a signal from soil (this fact explains item a), and they saturate for denser canopies (explains item b). In fact, due to variations in the optical properties of leaves and soils and different canopy structures, it is unclear if the same value of VIs corresponds to the same amount of "greenness" of different species, while LAI is directly comparable across species. Still, EVI2 is more robust than NDVI for phenology retrievals according to the above two criteria.

Next, focus on the seasonal profile of LAI only (c.f. Figure 6). The following should be noted: (1) there exists a large variability of seasonal profiles of species, mostly by amplitude but also by phase and duration of the growing season; (2) there is a good separability of seasonal profiles by forest classes (DBF have the largest seasonal max, followed by ENF and by DNF); (3) however, regardless of the class, LAI drops to low values outside the growing season even for ENF. Regarding item (1), all species have a single maximum except for stone birch, which exhibits a secondary small maximum early in the spring, potentially due to early understory development. Regarding item (3), the seasonality of ENF remains an issue under debate in terms of theoretical understanding [18] and in situ assessment due to insufficient temporal coverage of ground measurements [30]. The effect of seasonality over ENF is suppressed by VIs and becomes highly noticeable when using LAI; the values are

very low off the growing season, indicating that leaves (needles) are effectively turned off in the process of photosynthesis, potentially due to low ground temperatures.

To conclude, we propose a metric to rank different variables according to their utility to capture the seasonal dynamic of vegetation "greenness" and thus suitability for phenometrics retrievals. Figure 7 shows the seasonal minimum, maximum, and amplitude (max–min) of NDVI, EVI2, and LAI. As variables have different scales of dynamic range, we constructed cross-variables ratio, amplitude/max. This quantity effectively highlights a portion of dynamic ranges utilized in seasonal variations and available for retrievals of phenometrics. The average ratio for the analyzed variables is as follows: NDVI = 0.4, EVI2 = 0.6, and LAI = 0.9.

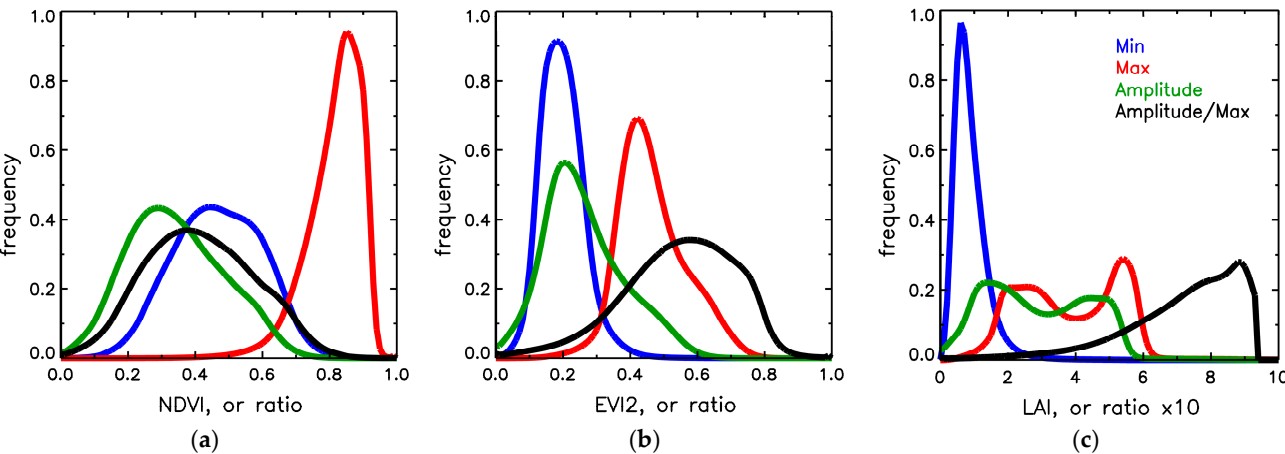

**Figure 7.** Comparison of variations of NDVI (**a**), EVI2 (**b**), and LAI (**c**) during the greenup phase. Each panel shows are histograms of min, max, and amplitude of corresponding variable. To intercompare utility of variables for phenology monitoring, amplitude/max ratio was also computed, quantifying portion of the total dynamic range [0, max] utilized over the course of seasonal variations.

### 3.3. Analysis of Retrieved Phenometrics

Below, we review features of key phenometrics retrieved from seasonal LAI profiles. The spatial distribution and histograms of max/min LAI and the day of reaching max/min LAI are shown in Figure 8. The distribution of $LAI_{max}$ essentially covers the full dynamic range of LAI and exhibits strong spatial variations. Two peaks at about LAI = 2 and LAI = 5.5 corresponds to low values in the Eastern Siberia, occupied by larch forests, and high values in the European and southern boundaries (Kaukaz), especially the Far East (Kamchatka Peninsula and Primorsky Krai), occupied by the broadleaved and some needleleaf forest species.

This high contrast between low and high values results in the mid-range values ($LA_{max}$~4) being suppressed. The histogram of maximum is not equivalent to any single daily histogram during summer (Figure 4), as there is about a 1-month lag (180–210) in the DOYs of reaching maximum value over the whole territory of Russian forests. The spatial distribution of days that reach the $LAI_{max}$ is not latitude but rather species dependent: later for DEF and ENF and earlier for DBF (cf. Table 2a and discussion below in this section). In terms of the distribution of $LAI_{min}$, there is a good spatial contrast between deciduous and evergreen forests. The day of reaching the minimum value seems latitude dependent.

Next, the statistical properties of key phenological dates of greenup and browning-down are considered, that is, DOYs of reaching 15%, 50%, and 90% amplitude during spring and fall periods as well as growing season duration. Figure 3a should be cross-referenced, which demonstrates the positioning of the above greening/browning dates over the course of LAI product availability: most dates are located within the highest level of data availability (~100%), except for spring at 15% greenup at the level of 70% of data coverage. The spatial distribution and histograms of phenological greening/browning dates and duration are presented in Figure 9. Moreover, 50% greening is reached on average at DOY = 157 (5 June), with a spread

between 15% and 90% amplitude, being 42 days. There is a clear latitudinal dependence on greening timing, increasing from South to North. A somewhat unexpected result is the earlier onset of spring in the North Siberian Lowland (NSL). Similar phenometrics were calculated for the browning period. On average, a 50% decrease in amplitude is reached at DOY 240 (27 August). The spread is ~50 days. Relatively earlier fall is especially noticeable in Eastern Siberia and the Sakhalin Peninsula. Finally, the distribution of the growing season duration is considered. On average, it is ~90 days. We can see quite a strong latitudinal dependence, i.e., a decrease from south to north, with the highest values observed in the Kaucaz and Primorsky Krai regions. The expected duration is low for all of the Eastern Siberia due to the central continent's location at high latitudes.

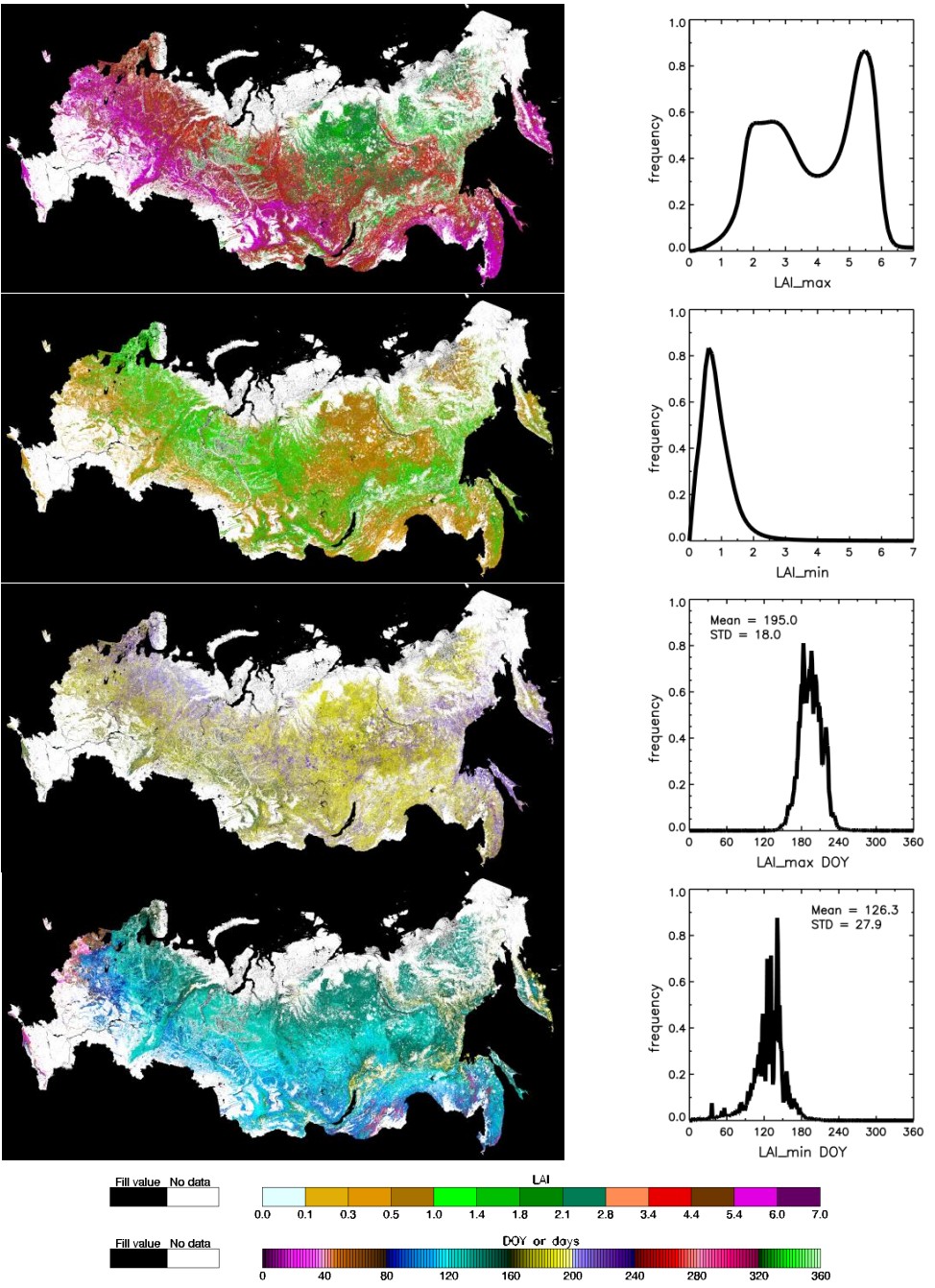

**Figure 8.** Mapping phenometrics of seasonal max/min of LAI over Russian forests. Spatial distributions and histograms of max/min LAI (first and second row) and DOY of reaching max/min (third and fourth rows) are shown. Maps are presented in the Albers projection at 230 m resolution.

**Table 2.** Statistics on phenometrics by forest species.

**(a)**

| Forest Classes | Forest Species | Greening Min LAI | | Max LAI | | Browning Min LAI | | LAI Amplitude | |
|---|---|---|---|---|---|---|---|---|---|
| | | LAI | DOY | LAI | DOY | LAI | DOY | Greening | Browning |
| DNF | Sparse larch | 0.9 | 145 | 2.3 | 195 | 0.5 | 271 | 1.4 | 1.8 |
| | Larch | 0.8 | 128 | 3.4 | 194 | 0.5 | 279 | 2.6 | 2.9 |
| ENF | Pine | 1.1 | 116 | 4.4 | 196 | 0.9 | 282 | 3.3 | 3.5 |
| | Spruce | 1.2 | 124 | 4.8 | 200 | 1.0 | 279 | 3.6 | 3.8 |
| | Fir | 1.3 | 118 | 5.0 | 192 | 1.1 | 270 | 3.7 | 3.9 |
| | Siberian pine | 1.1 | 111 | 4.8 | 196 | 0.9 | 283 | 3.7 | 3.9 |
| DBF | Oak | 0.5 | 79 | 5.9 | 183 | 0.4 | 318 | 5.4 | 5.5 |
| | Beech | 0.6 | 54 | 5.9 | 187 | 0.5 | 331 | 5.3 | 5.4 |
| | Stone birch | 1.1 | 160 | 5.5 | 207 | 0.5 | 294 | 4.4 | 5.0 |
| | Birch | 0.6 | 109 | 5.4 | 189 | 0.5 | 287 | 4.8 | 4.9 |
| | Aspen | 0.6 | 103 | 5.7 | 183 | 0.6 | 288 | 5.1 | 5.1 |
| | Linden | 0.5 | 104 | 5.9 | 182 | 0.4 | 309 | 5.4 | 5.5 |
| | Maple | 0.5 | 112 | 5.8 | 172 | 0.4 | 302 | 5.3 | 5.4 |

**(b)**

| Forest Classes | Forest Species | Greening dates and spread | | | | Browning dates and spread | | | | Duration days | | |
|---|---|---|---|---|---|---|---|---|---|---|---|---|
| | | 15% | 50% | 90% | Spread | 90% | 50% | 15% | Spread | 15% | 50% | 90% |
| DNF | Sparse larch | 151 | 165 | 185 | 34 | 209 | 235 | 253 | 44 | 102 | 70 | 24 |
| | Larch | 141 | 159 | 182 | 41 | 209 | 237 | 256 | 47 | 114 | 78 | 27 |
| ENF | Pine | 131 | 154 | 181 | 50 | 213 | 242 | 265 | 52 | 133 | 87 | 32 |
| | Spruce | 137 | 160 | 185 | 48 | 215 | 242 | 263 | 48 | 126 | 82 | 31 |
| | Fir | 127 | 145 | 171 | 44 | 214 | 241 | 259 | 45 | 132 | 96 | 43 |
| | Siberian pine | 126 | 150 | 178 | 52 | 214 | 243 | 265 | 51 | 139 | 92 | 36 |
| DBF | Oak | 121 | 138 | 159 | 37 | 212 | 252 | 278 | 66 | 156 | 114 | 53 |
| | Beech | 108 | 127 | 150 | 42 | 235 | 276 | 297 | 62 | 188 | 149 | 85 |
| | Stone birch | 166 | 175 | 191 | 25 | 225 | 249 | 266 | 41 | 100 | 74 | 35 |
| | Birch | 129 | 149 | 169 | 40 | 211 | 243 | 265 | 54 | 135 | 94 | 41 |
| | Aspen | 124 | 142 | 162 | 38 | 209 | 243 | 266 | 57 | 142 | 101 | 47 |
| | Linden | 122 | 137 | 156 | 34 | 217 | 254 | 273 | 56 | 151 | 116 | 60 |
| | Maple | 124 | 135 | 150 | 26 | 204 | 239 | 267 | 63 | 143 | 103 | 54 |

The detailed information on dependence of phenometrics on forest species/calsses is presented in Table 2. Seasonal $LAI_{max}$ values increase from DNF to ENF to DBF species. Seasonal $LAI_{min}$ is the lowest in DBF species. Seasonal $LAI_{min}$ of ENF is higher (0.9–1.3) than for other forest classes. In terms of the extent of the growing season, the duration is the longest for DBF (73–149 days), which comes from an earlier onset of greenness and later browning. The shortest duration (70–77 days) is observed for DNF species, which is explained by harsh weather conditions in the Eastern Siberia.

Next, we evaluated the impact of the choice of the base variable on the estimation of phenometrics. Specifically, we compared estimates of DOY of mid-greening, Mid-Browning, and duration from IKI LAI phenometrics data and NASA MODIS MCD12Q2 product (Figure 10). On average, using LAI results in the later onset of spring by 9.2 +/− 11.3 days and earlier onset of fall by 8.85 +/− 11.57 days. Altogether, this results in a shorter duration of the growing season by 19.89 +/− 17.95 days. Qualitatively, those are expected results according to theoretical considerations (Section 2.3). The spatial distribution of differences in spring and fall are "summarized" in the duration: the largest decreases (−30 days) are noticed in the northern region of the European part and the southern region of the Asian part of Russia. This excludes Primorsky Krai, where small (−10 days) decreases or even

small increases (+5 days) are observed. A small decrease is observed over the northern region of the Eastern Siberia. A potential reason for the small increase is the residual difference in seasonal profile processing (data interpolation) between algorithms.

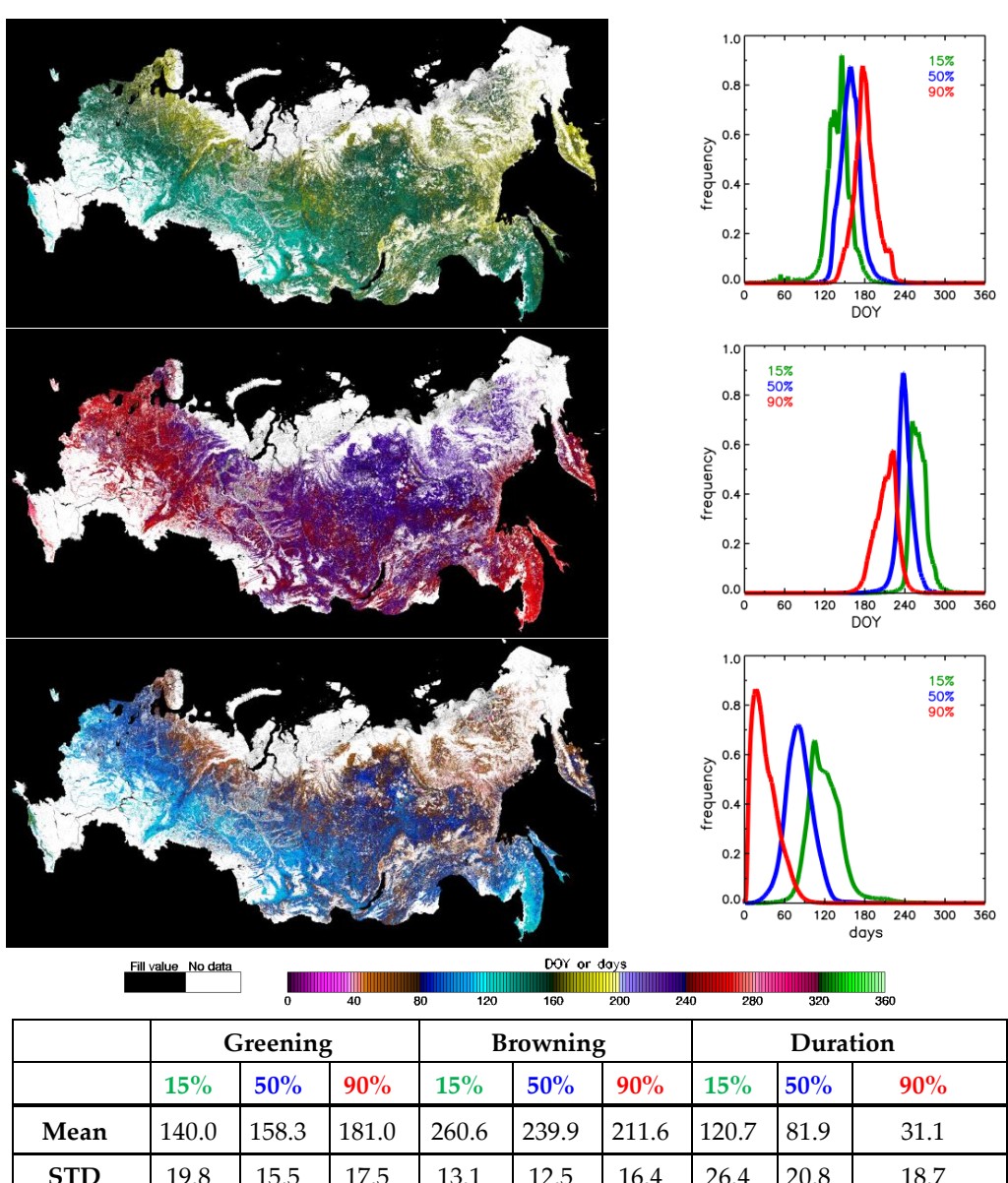

|  | Greening | | | Browning | | | Duration | | |
|---|---|---|---|---|---|---|---|---|---|
|  | **15%** | **50%** | **90%** | **15%** | **50%** | **90%** | **15%** | **50%** | **90%** |
| **Mean** | 140.0 | 158.3 | 181.0 | 260.6 | 239.9 | 211.6 | 120.7 | 81.9 | 31.1 |
| **STD** | 19.8 | 15.5 | 17.5 | 13.1 | 12.5 | 16.4 | 26.4 | 20.8 | 18.7 |

**Figure 9.** Mapping phenometrics (c.f. Table 1 for definition) of greening and browning phases of seasonal LAI variations over Russian forests. Spatial distribution and histograms for mid-greening, mid-browning, and duration are shown. To highlight the spread of greening/browning phases, histograms also include the distribution of DOYs when reaching 15% and 90% of amplitude, corresponding to greenup and maturity (dormancy and senescence). 15% and 90% histograms at the duration plot show the distribution of difference between dormancy and greenup DOYs and senescence and maturity DOYs. Maps are presented in the Albers projection at 230 m resolution.

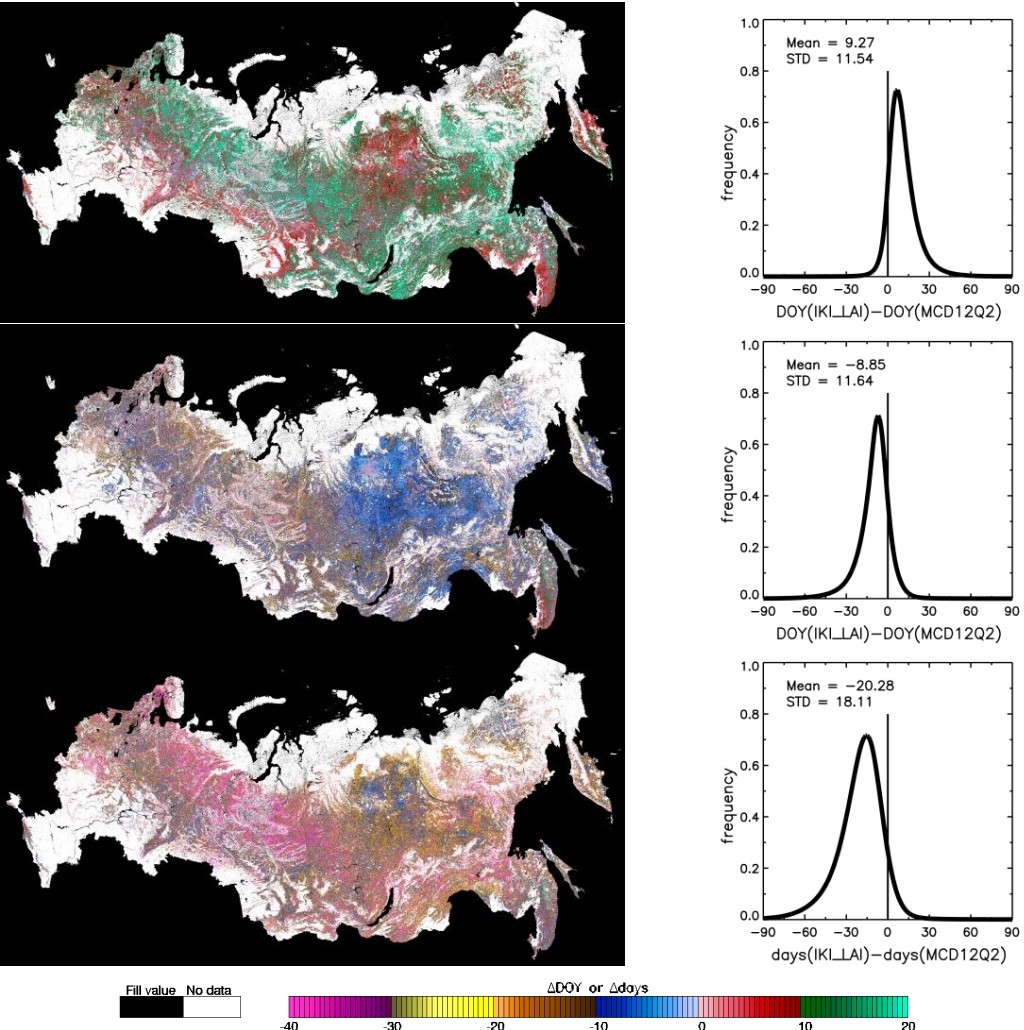

**Figure 10.** Comparison of phenometrics estimates from IKI MODIS LAI phenology and NASA MODIS MCD12Q2 products. The latter product is based on the seasonal EVI2 profiles. Spatial distribution and histogram are shown for mid-greening (**top row**) DOYs, mid-browning (**middle row**) DOYs, and duration (**bottom row**) days. Maps are presented in the Albers projection at 230 m resolution.

*3.4. Sensitivity Analysis*

Additional insights into forest functioning can be achieved when statistics on phenometrics are combined with those of the relationship between them and their drivers. One of the key drivers is solar illumination, which changes with latitude. To test the relationships, we implemented linear regressions. We need to emphasize that those regressions are very simplistic tests of complex environmental relationships with many forcing factors involved, resulting in a limited level of correlation with a single factor.

Figure 11 shows the results of the correlation between the duration of the growing season and greening/browning spreads. This relationship is an indicator of the inertia in the development of foliage. Regressions based on all forest pixels indicate that longer duration requires longer intervals to reach (Figure 11a,b). Also, duration is more sensitive to the browning spread ($\partial$duration/$\partial$spread = 0.24) than the greening spread ($\partial$duration/$\partial$spread = 0.18). When averaging duration and spread by species, one can notice (c.f. Figure 11c–e) that (1) spread is always lower than duration, (2) duration increases from DNF to ENF to DBF, while spread variability is much lower, (3) browning spread is always higher than greening spread, and the highest discrepancy occurs for DBF. The last phenomenon may have both natural causes (the physiological mechanism of the

development of leaves and degradation is different) and artificial (MODIS data stripes occur in fall only).

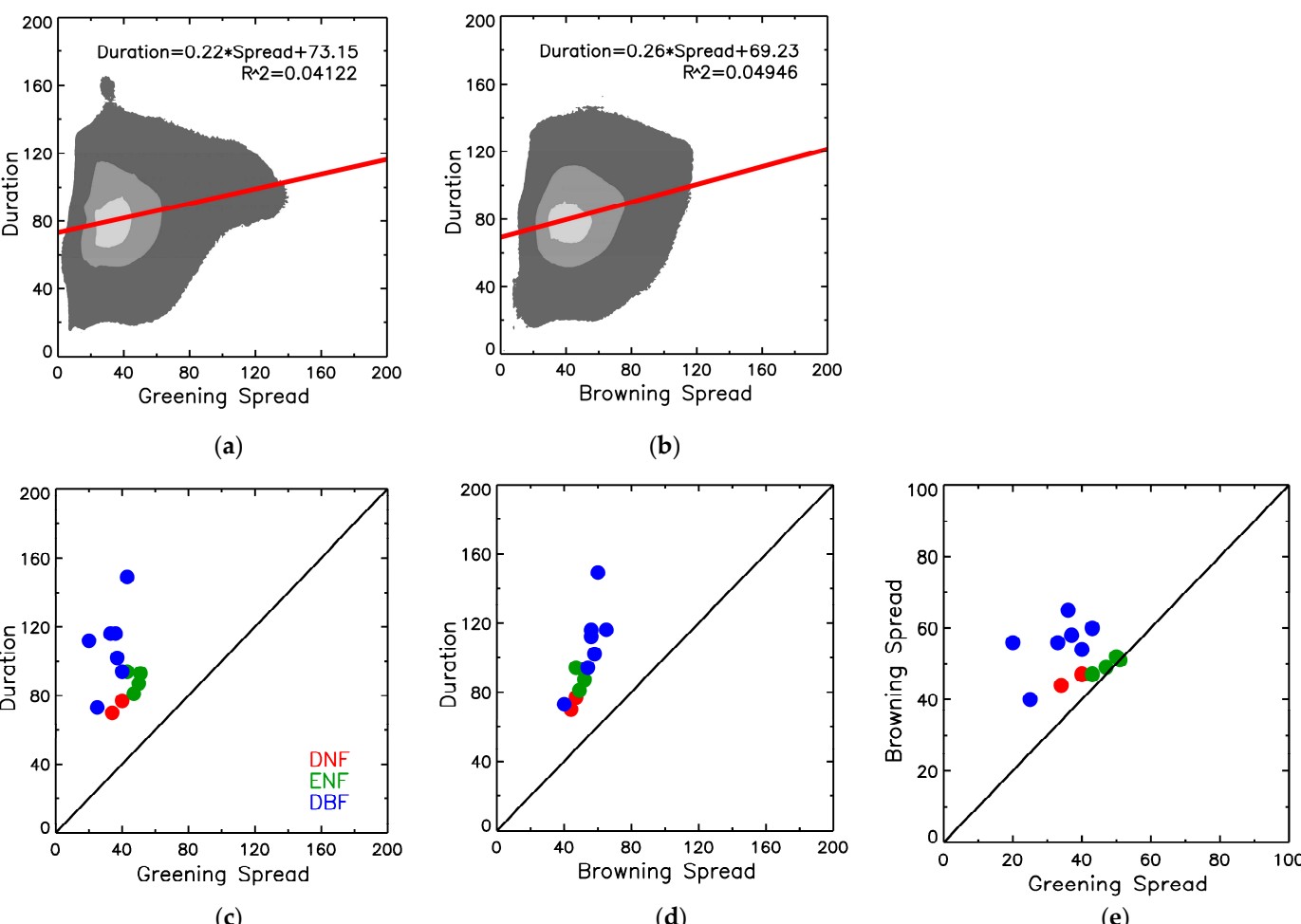

**Figure 11.** Statistical testing of the relationship between spread and duration (c.f. Table 1). Panel (**a**) shows data density (marked with shades of gray), linear regression lines (shown in red) and corresponding regression statistics for duration vs. greening spread, while panel (**b**) for duration vs. browning spread. Panels (**c**,**d**) show scatterplot of the same quantities averaged over individual species of forest classes (ENF, DNF, and DBF). Panel (**e**) demonstrates the relationship between averaged greening and browning spreads.

Next, we analyze the dependence of phenometrics on latitude, a proxy for solar illumination (cf. Figure 12). There is a clearly observable tendency of increasing $LAI_{max}$ from North to South ($\partial LAT / \partial LAI = -1.83$) with the largest (among other relationships) $R^2 = 0.24$ (Figure 12a). However, the day of reaching $LAI_{max}$ has virtually no relationship with latitude ($\partial LAT / \partial day = 0.01$) (Figure 12b). Different vegetation may respond differently to varying amounts of solar illumination, but solar illumination has the same seasonal course centered on DOY 172 (21 June) regardless of latitude. The fair question is why seasonal $LAI_{max}$ occurs not on 172 but later (between DOYs 174–207, depending on species, c.f. Table 2b)? We may speculate that the reason is foliage development inertia, a phenomenon, which occurs when foliage continues to develop while solar illumination forcing starts to decline.

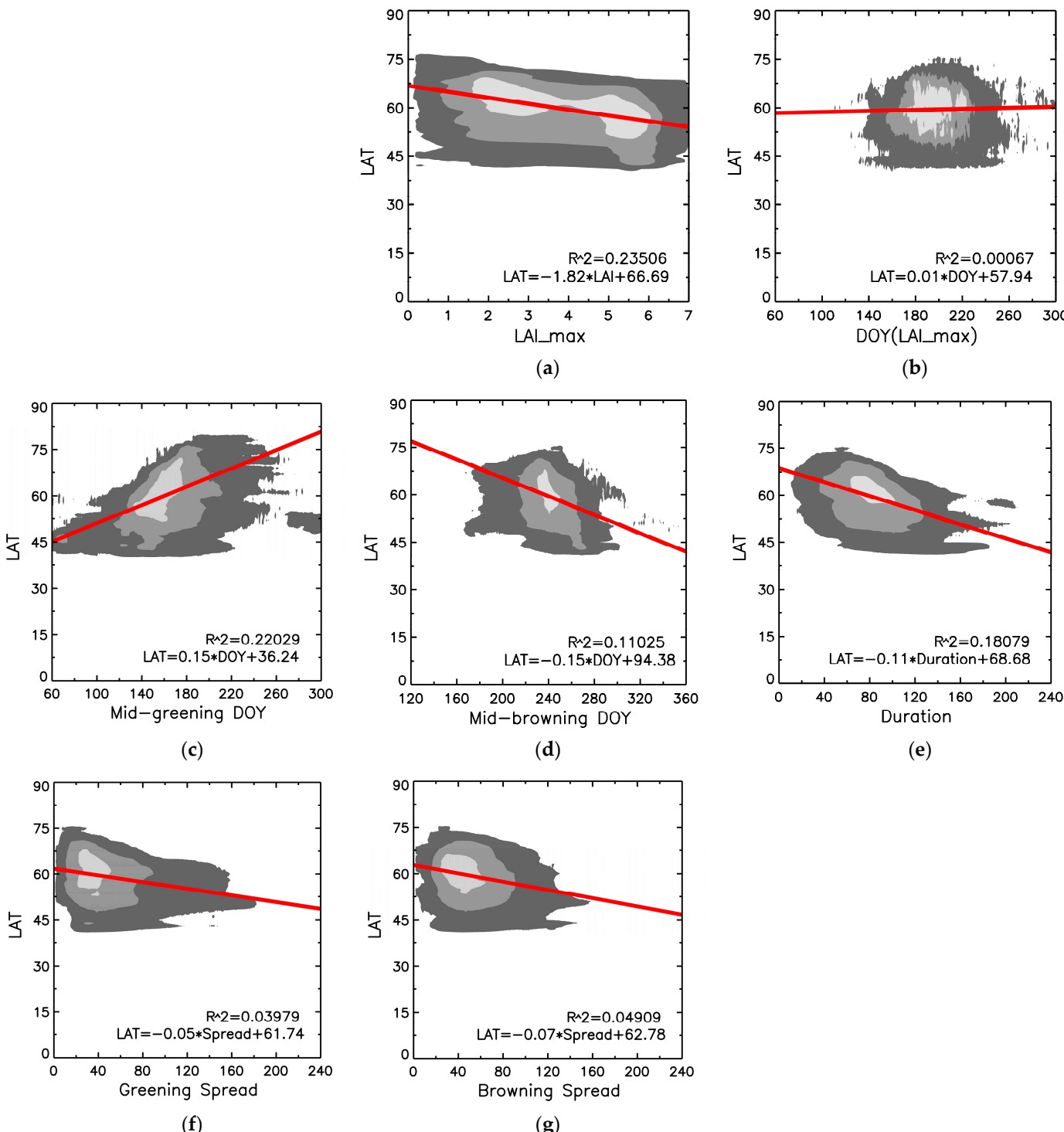

**Figure 12.** Statistical testing of the latitudinal dependence of various phenometrics: LAI_max (**a**), DOY of reaching LAI_max (**b**), mid-greening (**c**), mid-browning (**d**), duration (**e**), greening spread (**f**), and browning spread (**g**). Each panel shows data density (marked with shades of gray), linear regression lines (shown in red) and corresponding regression statistics.

Dates of mid-greening and mid-browning and duration also have substantial dependence on latitude (Figure 12c–e). Delayed spring, earlier fall, and consequently shorter duration occur as we move from south to north. Interestingly, the dependence of greening and browning spreads on latitude: both increase from south to north (Figure 12f,g).

This apparently expresses the fact that vegetation poses more inertia in the north due to weaker forcing.

## 4. Conclusions

In this study, we have demonstrated the advantages of LAI over other widely used variables, such as NDVI, FPAR, and EVI2, for phenometrics retrievals over Russian forests. Biophysical parameters LAI is sensitive to green vegetation only, while radiometric parameters VIs are only proxy to LAI as they are also sensitive to soil/background reflectance over sparse vegetation, they saturate over dense foliage, etc. The above properties of variables impact their seasonal course of variations: LAI better utilizes its dynamic range, drops to low values during winter, and is not affected by background reflectance outside the growing season. LAI greenup and browning-down rates are much steeper, resulting in a later onset of spring (9.3 days) and earlier onset of fall (8.9 days) compared to estimates based on EVI2 from the NASA MCD12Q2 product. Also, being a biophysical variable does not guarantee a better performance for phenology retrievals: FPAR saturates and has no substantial advantages over NDVI. In terms of dynamic properties for phenology retrievals, EVI2 is more robust than NDVI due to the lower impact of the above-mentioned limiting factors. In fact, after removing those limiting factors (using LAI), strong seasonality over ENF becomes apparent, the phenomenon, which is suppressed in VIs seasonal profiles. We attribute those not to needle fall but the suppression of photosynthesis during cold time. In this study, we have tabulated the key phenometrics of 12 species of Russian forests to address the needs of climate and ecological studies. While spatially varying, in general, the seasonal course of the Russian forest starts with min LAI at DOY = 124.1 +/− 27.3, reaches maximum at DOY = 194.3 +/− 18.0 and reaches back minimum at DOY = 281.4 +/− 17.7. Growing season duration is matched by snow cover and associated cloudiness, as expressed by the MODIS data availability seasonal curve. The key limitation on the accuracy of the derived phenometrics comes from the striping of MODIS data, which occurs in the fall and affects seasonal profiles both of VIs and LAI. Validation of retrieved phenometrics is equivalent to the validation of LAI seasonal profiles. Expansion of networks with automatic periodic digital camera measurements, such as Phenological Eyes Network http://www.pheno-eye.org (accessed on 13 November 2023), seems the most effective way to address the problem in the future.

**Supplementary Materials:** The following supporting information can be downloaded at: https://www.mdpi.com/article/10.3390/rs15225419/s1 [17,31–33].

**Author Contributions:** N.V.S. carried out the analysis and wrote the paper; V.A.E. and T.S.M. performed pre-processing of MODIS data; E.A.S. implemented GIS visualization of MODIS data; S.A.B. provided supervision, project administration, and funding acquisition. All authors have read and agreed to the published version of the manuscript.

**Funding:** This study has been performed under the national significance project, "Development of a system for ground-based and remote monitoring of carbon pools and greenhouse gas fluxes in the territory of the Russian Federation, ensuring the creation of recording data systems on the fluxes of climate-active substances and the carbon budget in forests and other terrestrial ecological systems" (state registration number: 123030300031-6). Data processing has been performed using the computing cluster "IKI-Monitoring" [34] supported through the "Monitoring" program (state registration number 122042500031-8).

**Data Availability Statement:** The data presented in this study are available on request from the corresponding author.

**Acknowledgments:** This work has been performed under a software release agreement with NASA Goddard Space Flight Center (GSFC). The authors thank S. Devadiga at NASA GSFC for providing the code of the NASA MODIS LAI algorithm (version/collection 6).

**Conflicts of Interest:** The authors declare no conflict of interest.

## Appendix A. IKI MODIS Forest Species Product

The IKI MODIS forest species is an annual product (from 2001 to present) at 230 m resolution, 12 classes of dominant tree species, refer to Figure 1 for a list of classes [35]. Input to the generation algorithm is a time series of weekly MODIS Red and NIR composites over the growing season as well as snow-covered ground composite generated with data over the January–April time frame. The classification algorithm is a Maximum Likelihood, implemented with the Locally Adaptive Global Mapping Algorithm (LAGMA) [36]. Training data sets include a digitized forest map of the USSR at a scale of 1:2.5 M and a time series (from 2001 to present) of the IKI MODIS landcover maps [35]. To serve as an input to the IKI MODIS LAI algorithm, the IKI forest species maps were combined with IKI land cover, to generate 8-biome IKI MODIS landcover, where forest species were aggregated to broader classes- ENF, DNF, and DBF (cf. Figure A1).

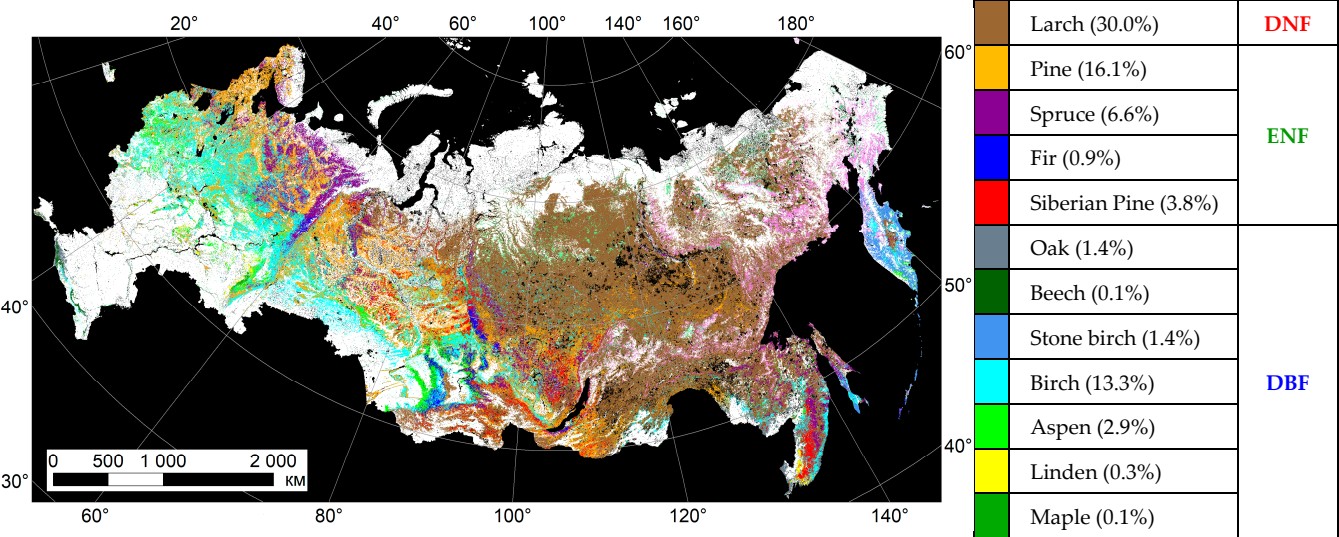

**Figure A1.** IKI MODIS dominant tree species of forest annual product for 2020, Percentage indicates portion of each forest species with respect to total forest area. ENF stands for Evergreen Needleleaf Forests, DNF is for Deciduous Needleleaf Forest, and DBF for Deciduous Broadleaf Forest. Map is presented in the Albers projection at 230 m.

## Appendix B. Striping Artefacts in MODIS Channel Data

Retrievals of phenometrics from MODIS data are significantly affected by one particular artifact of atmospheric correction of MODIS data- large-scale stripes in the Red channel. An example of this artifact is shown in Figure A2. The artifact appears as anomalously low reflectances at the Red Channel (DN < 100 on the scale of 0–$10^4$), the NIR channel is not affected, resulting in anomalously high values of retrieved LAI. The artifact usually appears at fall (DOY > 200) and low values of Solar Zenith Angle (SZA) < 600. This artifact is not screened by cloud mask, that is, it cannot be treated as cloud shadow, as there are no corresponding clouds. To minimize the impact, we screen MODIS data by the above thresholds on DN and SZA at the compositing step of data processing. However, this procedure also results in the removal of valid dark pixels (i.e., dark evergreen needleleaf forests). Further suppression of the artifact takes place at the interpolation step, especially considering that stripes are non-stationary. Nevertheless, stripes carry a residual artifact-appearing as a second peak in the fall on the seasonal LAI profile in addition to the summer seasonal peak.

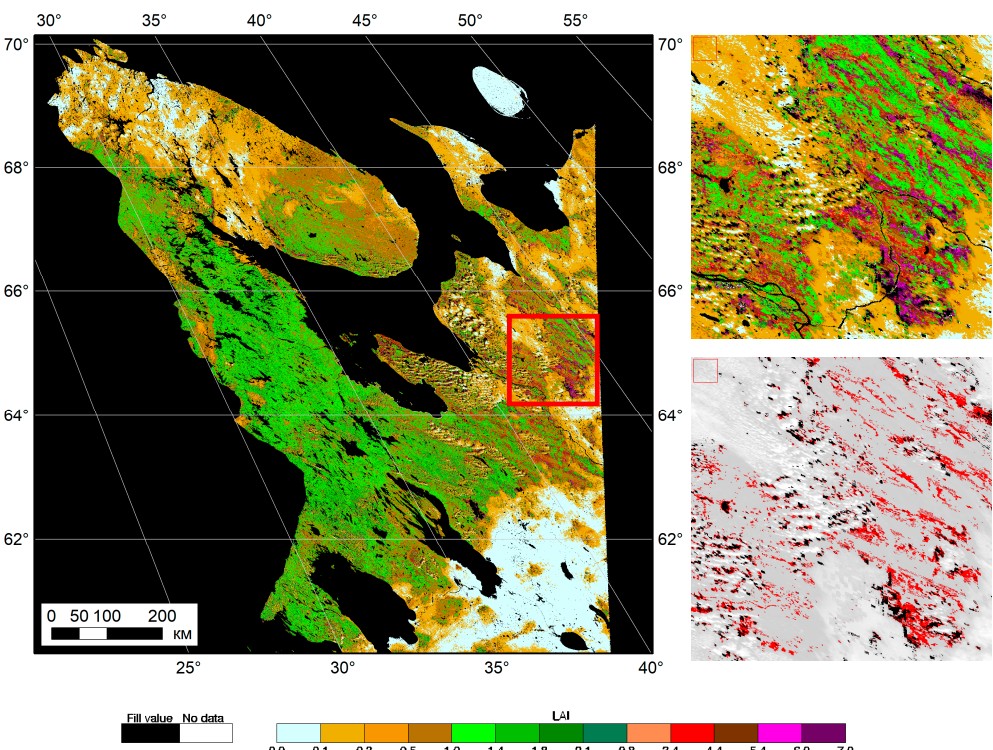

**Figure A2.** Striping artifacts in MODIS data. Shown (clockwise) are daily swath MODIS LAI data for MODIS tile h19v02, MODIS LAI, and Red channel subsets (marked with red frame in the previous image) for DOY 277 (3 October) in 2016. Daily LAI retrievals are performed for every pixel, regardless of cloud mask (cloud contamination is indicated by light turquoise color for very low LAI). MODIS Red channel data are shown in grayscale, red color mask (DN < 100) was applied to highlight potential areas affected by stripes. Maps are given in the sinusoidal projection at 230 m resolution.

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
