# Peer review of "Utility of Leaf Area Index for Monitoring Phenology of Russian Forests"

_remotesensing, doi:10.3390/rs15225419_

Round 1

Reviewer 1 Report

Comments and Suggestions for Authors

This manuscript demonstrated the advantages of LAI over other widely used parameters such as FPAR, NDVI and EVI2 for phenometrics retrievals over Russian forests. Meanwhile, the authors also did some sensitivity analysis for various phenometrics. There is a lot of great work in this manuscript. All figures are very pretty. This manuscript reads well and the structure of introduction and results section is logical. Overall, this manuscript is well written and can be considered for publication, but the manuscript can be improved. I think this manuscript can be accepted after a minor revision.

Minor revision:

1. I suggest that the last two paragraphs of section introduction should introduce your research purpose and significance and show highlights of your research.

2. Line 84 and 166, two “Data”? Please modify the title.

3. Please add full names of “DOY”, “DNF”, “ENF” and “DBF” when these words first appear in the manuscript.

4. Line 277, “…DOYs 120, 170, 200, 230, 280,….”, Why did you choose this five values?

5. Please add scale bars and north-arrows in your mapping figures.

6. Please improve the figure resolution, especially for Figure A2.

4. Please double-check references.

Author Response

Reviewer #1:

This manuscript demonstrated the advantages of LAI over other widely used parameters such as FPAR, NDVI and EVI2 for phenometrics retrievals over Russian forests. Meanwhile, the authors also did some sensitivity analysis for various phenometrics. There is a lot of great work in this manuscript. All figures are very pretty. This manuscript reads well and the structure of introduction and results section is logical. Overall, this manuscript is well written and can be considered for publication, but the manuscript can be improved. I think this manuscript can be accepted after a minor revision.

Minor revision:

  1. I suggest that the last two paragraphs of section introduction should introduce your research purpose and significance and show highlights of your research.

RESPONSE: The last two paragraphs of the Introduction exactly do what is requested: state the research purpose and highlights features of the research:

“In this study we have attempted both conceptually understand and practically assess utility of LAI with respect to other biophysical (FPAR) and radiometric (NDVI and EVI2) parameters for retrievals phenometrics over forests. The research has been implemented using MODIS data over full extent of Russian forests, which exhibit wide range of foliage density and its seasonal dynamics. Note, while several global (or Pan-Arctic) phenological products exist, we did not find focused in-depth assessment of phenology of this region.

This paper is organized as follows. Section 2 describes remote sensing products utilized in this research. Section 3 provides background for conceptual understanding of the difference in seasonal variations of base variables selected for this study: NDVI, EVI2, FPAR and LAI. We also briefly highlight features of our phenometrics retrieval algorithm. Section 4 reports on the results of this study, which includes analysis of data coverage limitations, comparison of dynamic properties of base variables, analysis of retrieved phenometrics and sensitivity analysis.”

  1. Line 84 and 166, two “Data”? Please modify the title.

RESPONSE: Thank you for pointing to this. The Editor also made suggestion to rename sections to standard journal headings “Materials and Methods”. All done.

  1. Please add full names of “DOY”, “DNF”, “ENF” and “DBF” when these words first appear in the manuscript.

RESPONSE: Done.

  1. Line 277, “…DOYs 120, 170, 200, 230, 280,….”, Why did you choose this five values?

RESPONSE: In fact, choice of dates was explained in the text “… for DOYs 120, 170, 200, 230, 280, covering major phases of seasonal changes from greening to browning.” That is, we just wanted to illustrate how shape of histograms of different variables is changing through the year.

  1. Please add scale bars and north-arrows in your mapping figures.

RESPONSE: Paper contains 11 maps. They were made small to fit format of the paper. Those maps have highly variable pattern. Thus adding lat/long grid and caption will make it difficult to comprehend spatial patterns and legend will be barely readable. While we are unable to add the requested features to each map, we have implement them for a single large map shown in the paper - forest species map (Fig. A1).

  1. Please improve the figure resolution, especially for Figure A2.

RESPONSE: The resolution of all plots was increased by factor of 8. Maps in Figure A2 were provided in the native resolution of MODIS tile (4800x4800 pix) and thus cannot be improved. However we adjusted color table for LAI to further highlight spatial variations.

  1. Please double-check references.

RESPONSE: Reference list was corrected. Also, according to the Editor request referencing style has been changed to fit journal standard (reference by paper number).

Reviewer 2 Report

Comments and Suggestions for Authors

Article looks decent and lengthy. It looks like there is a lot of work behind it, and authors have strong record in the field of research, so I would recommend publication of the article.

Author Response

No changes requested. Thank you for review.

Reviewer 3 Report

Comments and Suggestions for Authors

In the manuscript "Utility of LAI for Monitoring Phenology of Russian Forests," Shabanov et al. investigated the retrievals of Land Surface Phenology (LSP) metrics over Russian forests by comparing the utility of LAI and other biophysical variables (FPAR), and radiometric parameters (NDVI and EVI2) for phenometrics retrievals, and found that LAI exhibits better utilization of its dynamic range during the course of seasonal variations and better sensitivity to the actual foliage “greenness” changes and its dependence on forest species. This study emphasized the importance of LAI-based retrievals of LSP on the estimation of the duration of the growing season. The description of the method and results are thorough and well-written. I recommend publishing this paper after doing minor edits.

Comments:

1.     Line 66 “Wang et al (2018)” should be “Wang et al. (2018)

2.     Line 117-118 Please explain the abbreviations as your first used, such as ENF and DNF.

3.     Line 147 Give the references for those values

4.     Line 166 both title is data, please add more information to distinguish them.

5.     Figure 3. Add the legends of those color bar. Since you show the seasonal LAI information, it better to add the LAI values in Figure3(a).

6.     Line 280-281. It’s better to add the specific DOY for the beginning and end of growing season.

7.     Figure 4. Please add the labels (i.e., a, b, c, …) in this figure, and use those labels to illustrate your results.

8.     Line 290-292 Please add the figure label in those statements

9.     For the paragraphs” Still VIs exhibit deficiencies over LAI…” and “focus on the seasonal profile of LAI only...” Please add the figure label in your statements.

Author Response

Reviewer #3:

In the manuscript "Utility of LAI for Monitoring Phenology of Russian Forests," Shabanov et al. investigated the retrievals of Land Surface Phenology (LSP) metrics over Russian forests by comparing the utility of LAI and other biophysical variables (FPAR), and radiometric parameters (NDVI and EVI2) for phenometrics retrievals, and found that LAI exhibits better utilization of its dynamic range during the course of seasonal variations and better sensitivity to the actual foliage “greenness” changes and its dependence on forest species. This study emphasized the importance of LAI-based retrievals of LSP on the estimation of the duration of the growing season. The description of the method and results are thorough and well-written. I recommend publishing this paper after doing minor edits.

Comments:

  1. Line 66 “Wang et al (2018)” should be “Wang et al. (2018)”

RESPONSE: Thank you for pointing to this. The Editor also made suggestion to change references according to the journal standard (numerical references). We’ve implemented the Editor’s request. 

  1. Line 117-118 Please explain the abbreviations as your first used, such as ENF and DNF.

RESPONSE: we rewrote this sentence without abbreviations. Later in the text we introduced 3 forest classes (DNF, ENF and DBF)) and expanded each abbreviation.

  1. Line 147 Give the references for those values

RESPONSE: Reference to Jiang et al. (2008) was included

  1. Line 166 both title is data, please add more information to distinguish them.

RESPONSE: Thank you for pointing to this. The Editor also made suggestion to rename sections to standard journal headings “Materials and Methods”. All done.

  1. Figure 3. Add the legends of those color bar. Since you show the seasonal LAI information, it better to add the LAI values in Figure3(a).

RESPONSE: Legend is already given in the Figure caption, ie “… also shown are

intervals of DOYs when LAI reaches (or exceeds) the following portions of amplitude of variations: 90% over DOYs [180-211] shown in red,…” The meaning is quite complex, this is why we used not color table, but rather explanation of color coding in words. For better visual perception we changed color of the words (red, blue, green brown). 

LAI values are not shown in this plot: shown area DOYs where reaching 15%, 50%, 90% of amplitude of variations.

  1. Line 280-281. It’s better to add the specific DOY for the beginning and end of growing season.

RESPONSE: We added average values of minimum and maximum DOYs to the text

  1. Figure 4. Please add the labels (i.e., a, b, c, …) in this figure, and use those labels to illustrate your results.

RESPONSE: Conceptually this figure is a table with each panel carrying the same type of information (histogram of variables), where rows correspond to variables (NDVI, FPAR, EVI2, LAI) and columns to dates (120…280). It is most convenient to refer to individual panels as “LAI for DOY 170”, rather than “figure 4y”, because reference carry the meaning of the plot and there are too many references (4x5=20 panels). In Figure 4 lables for lines (variables names) are missing, which has been corrected.

  1. Line 290-292 Please add the figure label in those statements

RESPONSE: References to specific panels in Fig. 4 were added.

  1. For the paragraphs” Still VIs exhibit deficiencies over LAI…” and “focus on the seasonal profile of LAI only...” Please add the figure label in your statements.

RESPONSE: References to corresponding figures were added.